# A De Novo *DNM1L* Mutation in Twins with Variable Symptoms, Including Paraparesis and Optic Neuropathy

**DOI:** 10.3390/biom15091230

**Published:** 2025-08-26

**Authors:** Alessia Nasca, Alessia Catania, Andrea Legati, Rossella Izzo, Carola D’onofrio, Teresa Ciavattini, Eleonora Lamantea, Costanza Lamperti, Daniele Ghezzi

**Affiliations:** 1Unit of Medical Genetics and Neurogenetics, Fondazione IRCCS Istituto Neurologico Carlo Besta, 20126 Milan, Italy; alessia.catania@istituto-besta.it (A.C.); andrea.legati@istituto-besta.it (A.L.); rossella.izzo@istituto-besta.it (R.I.); caroladonofrio98@gmail.com (C.D.); teresa.ciavattini@utc.fr (T.C.); eleonora.lamantea@istituto-besta.it (E.L.); costanza.lamperti@istituto-besta.it (C.L.); daniele.ghezzi@istituto-besta.it (D.G.); 2Department of Pathophysiology and Transplantation (DEPT), University of Milan, 20122 Milan, Italy

**Keywords:** *DNM1L*, mitochondrial disorders, mitochondrial dynamics, mitochondrial and peroxisomes fission, variant reclassification

## Abstract

Mitochondrial network dynamics, encompassing processes like fission, fusion, and mitophagy, are crucial for mitochondrial function and overall cellular health. Dysregulation of these processes has been linked to various human diseases. Particularly, pathogenic variants in the gene *DNM1L* can lead to a broad range of clinical phenotypes, ranging from isolated optic atrophy to severe neurological conditions. *DNM1L* encodes DRP1 (dynamin-1-like protein), which is a key player in mitochondrial and peroxisomal fission. This study describes two twin sisters with a de novo heterozygous variant in *DNM1L*, due to possible paternal germline mosaicism identified through clinical exome sequencing. The two twins showed a variable clinical presentation, including paraparesis and optic neuropathy. Functional studies of patient-derived fibroblasts revealed altered mitochondrial and peroxisomal morphology, along with dysregulated *DNM1L* transcript levels, indicating the deleterious effect of the variant. These findings allowed us to reclassify the identified variant from a variant of uncertain significance to a likely pathogenic variant. Our report provides insight into the phenotypic spectrum of *DNM1L*-related disorders and highlights the need to combine genetic and functional analyses to accurately diagnose rare mitochondrial diseases.

## 1. Introduction

Mitochondria are membrane-bound organelles commonly referred to as the “powerhouses” of the cell due to their essential role in cellular energy production through adenosine triphosphate (ATP) via oxidative phosphorylation (OXPHOS). Beyond energy metabolism, they are also involved in crucial cellular processes such as apoptosis, calcium signaling, and the regulation of the cell cycle. These organelles are highly dynamic, continually undergoing fusions, fissions, and mitophagy to regulate their morphology and control their number and size, a process collectively referred to as mitochondrial network dynamics [1]. This dynamic is essential not only for bioenergetic efficiency but also for calcium homeostasis and redox signaling, ensuring mitochondrial quality control through segregation and removal of damaged organelles [2,3]. Accumulating evidence has emphasized the importance of mitochondrial network dynamics for mitochondrial function, and an imbalance of fission and fusion has been implicated in a variety of human diseases, including cardiovascular (cardiomyopathies, ischemia–reperfusion Injury) [4,5] and metabolic disorders (diabetes mellitus, obesity) [6], and neurodegenerative diseases (Alzheimer’s disease (AD), Parkinson’s disease (PD), Huntington’s disease (HD)) [7]. More recently, dysregulated mitochondrial dynamics have also been linked to immunological dysfunctions—such as aberrant NLRP3-inflammasome activation in autoinflammatory syndromes—and to cancer progression, where hyperfission supports the metabolic reprogramming and invasiveness of tumor cells [8,9]. Disorders of mitochondrial network dynamics are often caused by mutations in genes encoding proteins involved in core machinery components of fission (DRP1, FIS1, MFF, and MiD49/MiD51), fusion (MFN1, MFN2, and OPA1), and transport processes, mainly ensured by large GTPases belonging to the Dynamin family [2]. Diseases associated with these pathogenic variants can be inherited in an autosomal recessive or dominant manner, ranging in severity from isolated optic atrophy to lethal encephalopathy [10,11]. Dynamin-1-like protein (DRP1), encoded by *DNM1L*, is a key regulator of mitochondrial fission, a crucial process to generate new mitochondria and favor their transport and redistribution inside the cell; moreover, it facilitates the isolation of damaged organelles for degradation by mitophagy. DRP1 also has a role in the maintenance of peroxisomal morphology, shape, and function [12]. DRP1 is an evolutionarily conserved GTPase that contains an N-terminal GTPase head, a middle domain (MD), important for the tetramerization of the protein, a C-terminal GTPase effector domain (GED), and a non-conserved variable domain (VD) [13]. Structural studies have revealed how DRP1 assembles into cardiolipin-dependent concentric spirals and ring-like structures via self-homotetramerization at mitochondrial fission sites, where adaptor proteins such as FIS1 and MFF recruit it to initiate constriction and membrane scission via GTPase-dependent hydrolysis [14]. Inherited heterozygous *DNM1L* pathogenic variants can cause an autosomal dominant form of isolated optic atrophy [15] with decreased visual activity, dyschromatopsia, and central scotoma, with onset in the first to third decade of life. No other neurological signs were reported for this condition, and the corresponding variants have been supposed to act with a dominant-negative effect restricted to mitochondrial fission impairment [15]. Contrariwise, other heterozygous de novo pathogenic variants have been associated with severe neurological phenotype with neonatal or infantile-onset encephalopathy manifesting with hypotonia, weakness, psychomotor delay, drug-resistant seizures, microcephaly, sensory axonal neuropathy, and cardiomyopathy [16,17,18,19]. Moreover, a severe multisystem presentation can be caused by biallelic loss-of-function *DNM1L* variants [20]. The literature review indicates distinct correlations between higher brain function impairments and specific DRP1 protein domains: encephalopathy and developmental regression were commonly observed in patients with pathogenic variants in the MD and GED domains, while peripheral neuropathy, ataxia, dystonia, and spasticity were more frequently associated with variants in the GTPase domain. Interestingly, seizures, refractory status epilepticus (RSE), and development of cerebral atrophy were more frequent in patients with a pathogenic variant in the MD [21,22,23]. Laboratory findings of *DNM1L*-related disorders can vary widely: most of the patients do not have biochemical evidence of mitochondrial or peroxisomal dysfunction, and lactic acidosis and/or defects of the electron transport chain (ETC) have rarely been observed. Cultured fibroblasts from *DNM1L*-mutant subjects show filamentous, tangled, and hyperfused mitochondria with typical “chain-like” structures, and elongated peroxisomes [18,20,24]. Here, we describe the case of two twins presenting with an early-adult onset neurological impairment characterized by paraparesis (and optic neuropathy in one of them), both harboring a de novo variant in *DNM1L* in the GTPase domain. Being monozygotic, they provide a unique opportunity to explore phenotypic variability in individuals with the same genetic background, thereby highlighting the potential influence of additional genetic, epigenetic, or environmental modifiers.

## 2. Materials and Methods

### 2.1. Genetic Studies

Informed consent for genetic analysis was obtained from all family members involved in the study. Clinical Exome sequencing (ES) was performed on DNA from patient 1, using the TruSightOne panel (Illumina, San Diego, CA, USA). The resulting variants were analyzed using Expert Variant Interpreter (eVai) from enGenome as a prioritization tool, exploiting the Phenotype filter based on HPO terms.

Sanger sequencing was used to validate the relevant variants, as well as to check variant segregation in the family. To highlight possible germline mosaicism, we used the Illumina Nextera XT DNA Library Preparation Kit. American College of Medical Genetics and Genomics (ACMG) classification of new variants was carried out starting from those present in Varsome (https://varsome.com, accessed on 7 August 2025) and Franklin (https://franklin.genoox.com, accessed on 7 August 2025). Variant pathogenicity was predicted using the public tools SIFT, Polyphen-2, Mutation Taster, and CADD.

### 2.2. Cell Culture

Primary fibroblasts were obtained from a skin biopsy of patient 2.

Patient 1 did not consent to have a biopsy taken. Fibroblasts were cultured in high-glucose Dulbecco’s Modified Eagle Medium (Euroclone, Milan, Italy) supplemented with 10% fetal bovine serum (Euroclone, Milan, Italy), 1% penicillin/streptomycin (Euroclone), and 4 mM glutamine (Euroclone, Milan, Italy) at 37 °C in a 5% CO_2_ humidified atmosphere.

### 2.3. RNAseq Analysis

Total RNA was extracted from fibroblasts, and sequenced using the Illumina Stranded mRNA Prep, Ligation kit by 75 bp paired-end reads on the Illumina NextSeq 500 platform (Illumina, San Diego, CA, USA). Generated fastq files were used as input for the Salmon pipeline [25] to calculate counts per transcript per gene. Differential expression and pathway analysis were conducted using the web application iDEP2.01 [26]. The DESeq2 package was used for gene expression analysis, applying a cut-off of a false discovery rate (FDR) ≤ 0.05 and a log2 fold change of 2 [27]. Gene ontology (GO) biological process was used to identify the enrichment of significant pathways [28,29].

### 2.4. Immunoblot

SDS-PAGE was performed in fibroblast samples, and 50 μg of protein from fibroblast homogenate was loaded for each sample in a 12% denaturing SDS polyacrylamide gel electrophoresis. A polyclonal antibody against DNM1L/DRP1 (D6C7, #8570, Cell Signaling, Danvers, MA, USA), a monoclonal antibody against GAPDH (#MAB374, Millipore, Burlington, MA, USA), a monoclonal antibody against OPA1 (#312607, BD Biosciences, San Jose, CA, USA), a monoclonal antibody against MFN2 (#9482, Cell Signaling), and a monoclonal antibody against TUBULIN BETA (T0198, Sigma, St. Louis and Burlington, MA, USA) were used. The quantification was conducted using ImageJ (Fiji) software (1.53t version) [30].

### 2.5. Fluorescence Microscopy

The mitochondrial fluorescent dye MitoTracker Red-CMXRos (Invitrogen, Carlsbad, CA, USA) was added at a final concentration of 50 nM for 30 min to the culture medium, for visualization of the mitochondrial network, and then visualized by fluorescence microscopy in live. For peroxisomal immunostaining, after fixation and permeabilization, cells were incubated with an anti-PMP70 (ABT12, Millipore) antibody, followed by an Alexa Fluor 488 secondary antibody (Invitrogen). Images were acquired with a confocal microscope (Leica TSC-SP8, Leica Microsystems, Mannheim, Germany). Analysis of the mitochondrial network and peroxisomal morphology was conducted using the image processing package ImageJ (Fiji) software (1.53t version). To quantify the morphological alterations, we used the form factor (“circularity”), where a value of 1.0 indicates a perfect circle while values tending to 0.0 indicate increasingly elongated shapes.

## 3. Results

### 3.1. Clinical Features

The case describes two female twins born to healthy parents, with two older brothers reported in good health. Family history disclosed first-grade consanguinity in the parents; the family pedigree is shown in Figure 1A.

Patient 1 (Pt1) presented with a progressive decline in visual acuity in her late twenties and was subsequently diagnosed with “optic neuropathy plus.” In addition to a severe bilateral, asymmetric optic neuropathy, she exhibited longstanding psychiatric manifestations, including depressive psychosis and obsessive–compulsive behaviors, which were managed with antipsychotic medication. Over time, her neurological condition deteriorated: during follow-up (last clinical assessment at age 36), she developed a progressive gait disturbance that evolved into a clinically manifest spastic paraparesis. Neurological examination revealed increased lower limb muscle tone, exaggerated deep tendon reflexes, and bilateral ankle clonus. Her perinatal history was unremarkable, and developmental milestones, including motor skills acquisition, were reported as normal during early childhood. Genetic testing excluded common mitochondrial DNA mutations responsible for Leber hereditary optic neuropathy (LHON), as well as pathogenic variants in the OPA1 gene.

A twin sister, patient 2 (Pt2), presented with a slowly progressive gait disturbance beginning in her early thirties. The condition gradually evolved into mild spastic paraparesis. Unlike her sister, she did not report any visual impairment. Her medical history included normal motor development and recurrent episodes of muscle pain during adolescence. Electromyographic studies and spinal cord MRI did not reveal significant abnormalities. Neurological evaluation confirmed the spastic component of the gait disorder. Her visual acuity remained within normal limits, and a comprehensive neuro-ophthalmological examination excluded the presence of optic atrophy or other signs of optic neuropathy.

### 3.2. Genetic Analysis

Clinical Exome sequencing was performed in the proband Pt1. Bioinformatics analysis, coupled with phenotypic filtering using the HPO terms “Optic atrophy” (HP:0000648) and “Gait disturbance” (HP:0001288), led to prioritizing a variant in the *DNM1L* gene (NM_012062.5). The heterozygous variant c.121G > A p.Val41Met was confirmed by Sanger and found to be present in the twin sister Pt2. The variant was not present in gnomAD or other public SNP databases, affected a highly conserved residue (Conservation Scores PhyloP100: 9.623), and had high scores of pathogenicity, according to different bioinformatics tools (mean of in silico predictors indicates the ACMG criterion: PP3_Moderate). The ACMG classification was Variant of Uncertain Significance–VUS (criteria PM2, PP3, PM1, PP2). Given that *DNM1L* mutations may be associated with recessive or dominant traits, we evaluated the segregation in the available family members by Sanger sequencing: the variant c.121G > A was absent on the mother’s blood DNA, also tested with NGS-XT deep sequencing to highlight possible germline mosaicism. Also, among the two healthy brothers, the variant was not present. The DNA from the father, reported to be not affected, was not available. To overcome this limitation, we performed further studies on family member haplotypes.

A frequent SNP c.120A > C p.Ser40Ser (rs10844308, approximate minor allele frequency-MAF in gnomAD: 12.4%) [31] was present alongside the c.121G > A variant in the twins, and also in one of the healthy siblings, but not in their mother. Visual inspection on Integrative Genomics Viewer (IGV, version 2.16.0) [32] of the NGS reads showed that the two adjacent variants were on the same allele in the twins (Figure 1B).

Analysis of informative microsatellites showed that all subjects were offspring of the same father (not shown). In addition, to better define the *DNM1L* alleles, we also verified the segregation of other frequent SNPs found in the *DNM1L* gene: c.252G > A p.Gly84Gly (rs2272238), MAF 12.9%; c.918A > G p.Thr306Thr (rs10844318) MAF 12.4%; and the rarer c.1968C > T p.Leu656Leu (rs148634653) MAF 1.8%. The segregation of these SNPs allowed us to discriminate between the two paternal alleles and to confirm that the variant c.121G > A p.Val41Met was in cis in the allele with all the other SNPs, which was of paternal origin. The nucleotide change c.121G > A was likely a de novo event due to paternal germline mosaicism (Figure 1A).

Because of the reported consanguinity of the parents, we searched for homozygous rare variants in the proband, but no candidate gene/condition associated with a recessive trait and showing phenotypic overlap with the two twins was identified.

### 3.3. Immunostaining and Imaging in Fibroblasts

To elucidate the functional consequences of the newly identified *DNM1L* variant, we first examined DRP1 protein by immunoblot analysis on fibroblasts obtained from Pt2. We observed a normal level of DRP1, in contrast with the strong reduction present in *DNM1L*-recessive patient cases [13]. In particular, the Pt2 sample showed a significant increase in additional bands immunoreactive to DRP1-antibody, with molecular weights lower than the classical DRP1 doublet signal (Figure 2A,B).

To assess if dysregulation of DRP1 affects the balance of fusion markers, we also checked the steady state levels of the primary mitochondrial fusion mediators, proteins like OPA1 and MFN2, but the analysis did not reveal any significant alterations (Figure 2C,D). Next, we performed morphological studies on the patient’s fibroblasts to assess the role of DRP1 on the dynamics of mitochondria and peroxisomes. The mitochondrial network of *DNM1L*-mutant fibroblasts displayed an altered mitochondrial morphology, with swellings, dots, and “chain-like” structures (Figure 3A), evaluated with Mitotracker red staining. By immunovisualization using an antibody against a peroxisomal protein, PMP70, we observed larger and more elongated peroxisomes in the patient’s cells (Figure 3B). Morphometric analysis confirmed an altered mitochondrial conformation of the mitochondrial network in the patient compared to controls, with a significantly increased circularity shape factor due to dots and “chain structures” (Figure 3C). Similarly, significantly reduced circularity was found for the patient’s peroxisomes, corresponding to an aberrant and elongated peroxisomal shape (Figure 3D). Taken together, these results demonstrate that the *DNM1L* variant disrupts DRP1 processing, leading to defective fission of both mitochondria and peroxisomes and culminating in widespread alterations of organelle architecture without compensatory up-regulation of fusion proteins.

### 3.4. Transcriptomics Analysis

To further explore the molecular consequences of the identified *DNM1L* variant, we conducted a comprehensive transcriptomic analysis through RNA sequencing (RNA-seq), aiming to assess both the potential effects on *DNM1L* transcript structure and abundance, as well as the broader impact on global gene expression. The analysis compared fibroblasts from the patient (Pt2) to a cohort of eleven control fibroblast lines (Ct1–11), providing a robust baseline for differential expression analysis.

Visualization of RNA-seq read alignments using Sashimi plots revealed no evidence of aberrant splicing events across the *DNM1L* gene locus, suggesting that the variant does not interfere with canonical splicing mechanisms. Quantification of *DNM1L* transcript abundance, expressed in transcripts per million (TPM) relative to the reference gene ENSG00000087470.19, showed that Pt2 fibroblasts retained approximately 98% of *DNM1L* transcript levels compared to controls. These data are in agreement with the immunoblot analysis and suggestive of a dominant trait. Interestingly, when evaluating the TPMs for all individual *DNM1L* transcripts, we found a general perturbation of individual entity counts with a particular increase in counts of ENST entries mostly linked to “no protein” DRP1 short isoforms in the patient compared to other control fibroblasts (Appendix A). The presence of this alteration suggests that the identified variant may confer increased stability or altered processing to transcript species that are typically unstable or underexpressed in normal fibroblasts, pointing to a regulatory disruption that may contribute to the disease mechanism. To evaluate the overall effect of the variant on gene expression, we performed a principal component analysis (PCA). The results showed a clear separation between Pt2 and control samples, indicating a distinct transcriptomic signature in the patient. This distinction was particularly evident in the 3D-PCA plot based on the expression of significantly altered genes. Differential expression analysis using the DESeq2 algorithm identified a total of 757 differentially expressed genes (DEGs), with 198 genes significantly upregulated and 559 genes downregulated in the patient’s fibroblasts compared to controls. Functional enrichment analysis of these DEGs, based on Gene Ontology (GO) Biological Processes (BP), highlighted several disrupted cellular pathways. Notably, among the top ten enriched pathways based on downregulated DEGs, “Organelle fission” emerged as one of the most significantly affected categories (adjusted *p*-value = 6.98 × 10^−33^, fold change = 5), strongly supporting the functional relevance of the *DNM1L* variant and its impact on mitochondrial and peroxisomal dynamics (Figure 4A–E). Together, these transcriptomic data provide compelling evidence that the *DNM1L* variant exerts widespread effects on gene expression, particularly within pathways directly linked to organelle homeostasis and fission processes, consistent with the morphological and protein-level abnormalities observed in the patient’s fibroblasts.

## 4. Discussion

*DNM1L* pathogenic variants were initially linked to a lethal encephalopathy due to defective mitochondrial peroxisomal fission 1 (EMPF1, MIM#614388), but then *DNM1L*-related disorders (MIM*603850) have been associated with a phenotypic spectrum, including neonatal or infantile-onset encephalopathy with hypotonia, ataxia, peripheral neuropathy, varying degrees of epilepsy, and cognitive impairment with both recessive or dominant inheritance. Furthermore, single heterozygous variants have been identified in cases with nonsyndromic optic atrophy 5 (OPA5) (MIM#610708) [15,20,33,34]. An increasing number of reports about the dominant variants suggest that it is possible to distinguish the severity of the phenotype according to the region in which the variant falls. Variants in the GTPase domain appear to be associated with a milder phenotype, limited to isolated optic atrophy; disease-causing variants in the central domain have been reported to cause severe encephalopathy with epilepsy and/or developmental delay/regression, and very few cases report variants in the GED domain associated with less severe epilepsy, optic atrophy, and impaired mobility [11,35,36]. Often, the newly described variants are not supported by functional studies showing that the variant is truly deleterious. When feasible, functional analyses can be carried out using simplified experimental models such as yeast, which offer a robust system for investigating mitochondrial function and gene impact. These models enable comprehensive evaluation through assays, including oxidative growth, oxygen consumption, petite colony frequency, and assessment of mitochondrial network architecture [18,20,21,37,38]. However, their use may be limited when the affected amino acid residue is not conserved or lacks sequence homology with the model organism. In such cases, when available, functional studies are performed directly on fibroblasts derived from patients carrying pathogenic *DNM1L* variants, revealing significant alterations in mitochondrial morphology, peroxisome fission, and overall cellular metabolic function [18,20,39].

Recently, the same variant we identified in the two twins has been described as a VUS in another patient presenting with adolescent-onset sensory neuronopathy, spasticity, dystonia, and ataxia [40]. Clinical workup also showed bilateral pallor of the optic disks. Interestingly, both this case and ours were characterized by a dominant inheritance from a germline mosaic parent and a milder presentation than many other described *DNM1L* cases. However, unlike the above-mentioned case, our patients showed gait impairment with mild lower limbs spasticity (increased muscle tone, hyperreflexia, and ankle clonus) without signs of ataxia or extrapyramidal involvement. Some degree of clinical variability is also detectable within our family, as shown by a higher severity of the phenotype with psychiatric comorbidity in Pt1 but, above all, by the absence of manifest optic involvement in one of the sisters (Pt2), while optic atrophy was the first symptom in the other twin (Pt1). Phenotypic heterogeneity is common in human diseases caused by mutations in the same gene and can be at least partially attributable to specific individual genetic backgrounds and environmental factors. However, intrafamilial variability is much rarer and typically limited in monozygotic twins because of their common genotypes.

Levodopa treatment had a positive effect in two separate cases described in the literature on speech, walking, and paroxysmal dystonia [40,41]. This type of treatment could also be evaluated in the case of our twins. Given the role of dopaminergic modulation in spasticity, future therapeutic trials may assess the benefits of levodopa, taking cues from existing single-case data and potential neuroprotective effects via mitochondrial pathways.

A peculiar molecular finding in our pedigree was the presence of a de novo variant in two twins, strongly suggestive of parental mosaicism. Genetic investigations by NGS allow the identification of low-level variants present in a somatic mosaic state in biological samples like blood DNA; they may indicate their presence as a germline variant that can be inherited. The studies we performed on the mother’s DNA did not show any evidence of the variant, while the haplotype analysis suggested the variant is on the paternal allele. Unfortunately, the unavailable DNA from the father hampered a direct assessment of this hypothesis. Nonetheless, the inferred paternal germline mosaicism stresses the importance of deep-sequencing strategies and parental haplotyping for accurate recurrence risk estimation during genetic counseling in clinical care [42,43]. Our functional analyses performed on patient-derived fibroblasts provided crucial evidence to support the reclassification of the *DNM1L* variant from a Variant of Uncertain Significance (VUS), originally based on the criteria PM2, PP3, PM1, and PP2, to a “likely pathogenic” (LP) classification. This reassignment was made possible by incorporating the PS3_moderate criterion, which accounts for well-established functional studies showing a deleterious effect on gene function. Additionally, the presence of the same variant in two unrelated cases, as reported in the literature, allows for the application of PP5_supporting, further strengthening the new classification in accordance with ACMG guidelines. The experimental findings also provide insight into the potential pathogenic mechanism underlying the p.Val41Met substitution. We hypothesize that this variant may confer increased stability or altered processing to specific *DNM1L* transcript isoforms that are normally degraded or expressed at minimal levels in healthy cells. This could plausibly account for the detection of additional lower-molecular-weight DRP1-immunoreactive bands on Western blot, which may arise from translation of these otherwise cryptic or unstable transcripts. Alternatively, these aberrant bands might represent products of non-canonical degradation pathways acting on the mutant DRP1 protein, reflecting a broader dysregulation of proteostasis linked to the variant. Such molecular alterations are consistent with previously described dominant-negative effects of other *DNM1L* mutations, wherein defective DRP1 subunits interfere with proper oligomer formation, impairing mitochondrial fission and leading to abnormal organelle morphology [44]. The phenotypic consequences of these disruptions—observed in both mitochondrial and peroxisomal networks—underscore the critical role of DRP1 in maintaining organelle homeostasis. In conclusion, we report and functionally characterize a de novo *DNM1L* variant (p.Val41Met) in a pair of monozygotic twins, providing strong evidence for its likely pathogenicity. Clinically, the variant is primarily associated with optic neuropathy and progressive spastic paraparesis, although intrafamilial phenotypic variability is evident. These findings expand the mutational and clinical spectrum of *DNM1L*-related disorders and emphasize the importance of integrated functional genomics in variant interpretation.

## 5. Conclusions

Our report highlights the critical diagnostic value of integrating NGS sequencing technologies with functional studies using cellular models, particularly in the context of rare genetic diseases, in cases of novel disease genes, novel variants, or phenotypic expansions. In such scenarios, genetic data alone are often insufficient to establish pathogenicity with confidence, and the addition of functional assays becomes instrumental in reclassifying uncertain variants. In this study, the identified *DNM1L* variant—although previously described in the literature—would have remained classified as a variant of uncertain significance (VUS) without the support of functional validation. The cellular model-based experiments provided compelling evidence of the deleterious nature of this specific variant, ultimately allowing for a more definitive interpretation of its pathogenicity. This reinforces the importance of moving beyond purely computational or sequence-based analyses when interpreting rare variants. Furthermore, this report provides a better understanding of the impact of the *DNM1L* variant on phenotypic findings, which will update the clinical diagnosis of *DNM1L*-related diseases, potentially aiding clinicians in future diagnoses and in the refinement of genotype–phenotype correlations.

## Figures and Tables

**Figure 1 biomolecules-15-01230-f001:**
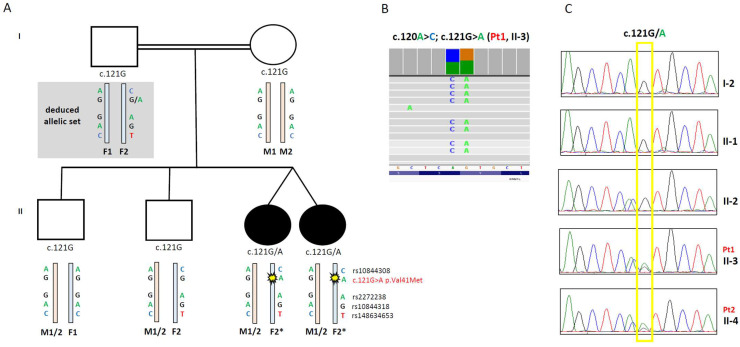
Family pedigree, alleles, and SNPs segregation. (**A**) Pedigree of the patient’s family, with analysis of *DNM1L* variant segregation and allele organization. (**B**) Snapshots from IGV software (Version 2.16.0) of the *DNM1L* variants identified in patient 1. The changes c.120A > C; c.121G > A were always present on the same reads, indicating they are on the same allele. (**C**) Electropherograms of the *DNM1L* regions containing the c.120A > C; c.121G > A variants in different family members.

**Figure 2 biomolecules-15-01230-f002:**
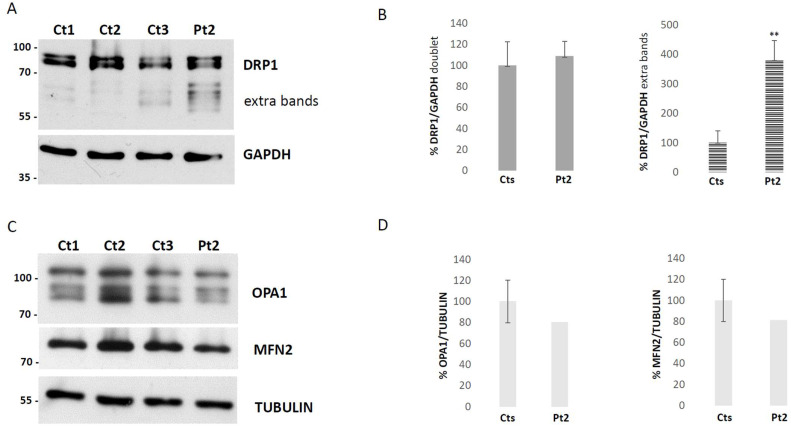
Immunoblot analysis on fibroblasts. Representative image of the Western blot analysis and densitometric quantification of DRP1, (**A**) DRP1 antibody immunoreactive bands from fibroblast homogenates of three controls (Ct 1-3) and patient 2 (Pt2); right, Western blot original images can be found in Appendix A. (**B**) graphical representation of the quantitative expression of the classical DRP1 doublet signal and the significant increase in extra bands in patient 2 compared to controls. The error bars indicate the standard deviation of samples in two different experiments. *t*-test (t(df) = 3) ** *p* < 0.01. Representative image of the Western blot analysis and densitometric quantification of fusion mitochondrial proteins like OPA1 and MFN2 to assess if dysregulation of DRP1 affects the balance of fusion markers, (**C**) antibodies immunoreactive bands from fibroblast homogenates of three controls (Ct 1–3) and patient 2 (Pt2); right, Western blot original images can be found in Appendix A. (**D**) graphical representation of the quantitative expression of OPA1 and MFN2 in patient 2 compared to controls. The error bars indicate the standard deviation of the three controls.

**Figure 3 biomolecules-15-01230-f003:**
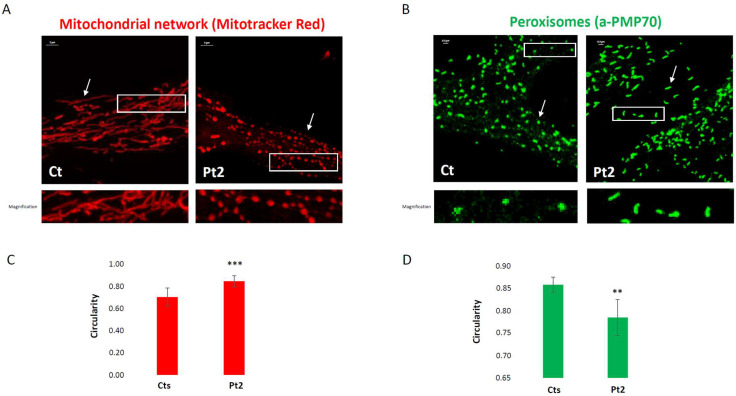
Morphological and morphometric analysis of the dynamics of mitochondria and peroxisomes in fibroblasts. Representative confocal images of fibroblasts from Ct and Pt2. (**A**) Fibroblasts were stained with Mitotracker Red dye to visualize the mitochondrial network. Scale bar: 5 μm. The white arrow points out morphological differences in shape between the groups of representative objects filamentous in the control or chain-like in the patient. The white rectangle indicates the digitally magnified area. (**B**) Peroxisomes were stained with a specific marker protein (PMP70). Scale bar: 0.5 μm. The white arrow points out morphological differences in shape between the groups of representative objects, dots in the control or dashes in the patient. The white rectangle indicates the digitally magnified area. (**C**,**D**) Graphical representation of the quantitative form factor “circularity”. The error bars indicate the standard deviation. A total of 44 field images from four healthy controls (10 for each) and 10 field images from patient 2 were used for the analysis of the mitochondrial network. (t(df) = 52) *** *p* < 0.001. A total of 2540 objects from three images of the healthy control and a total of 2844 objects from three images of patient 2 were used for the analysis of peroxisome morphology. (t(df) = 4) ** *p* < 0.01.

**Figure 4 biomolecules-15-01230-f004:**
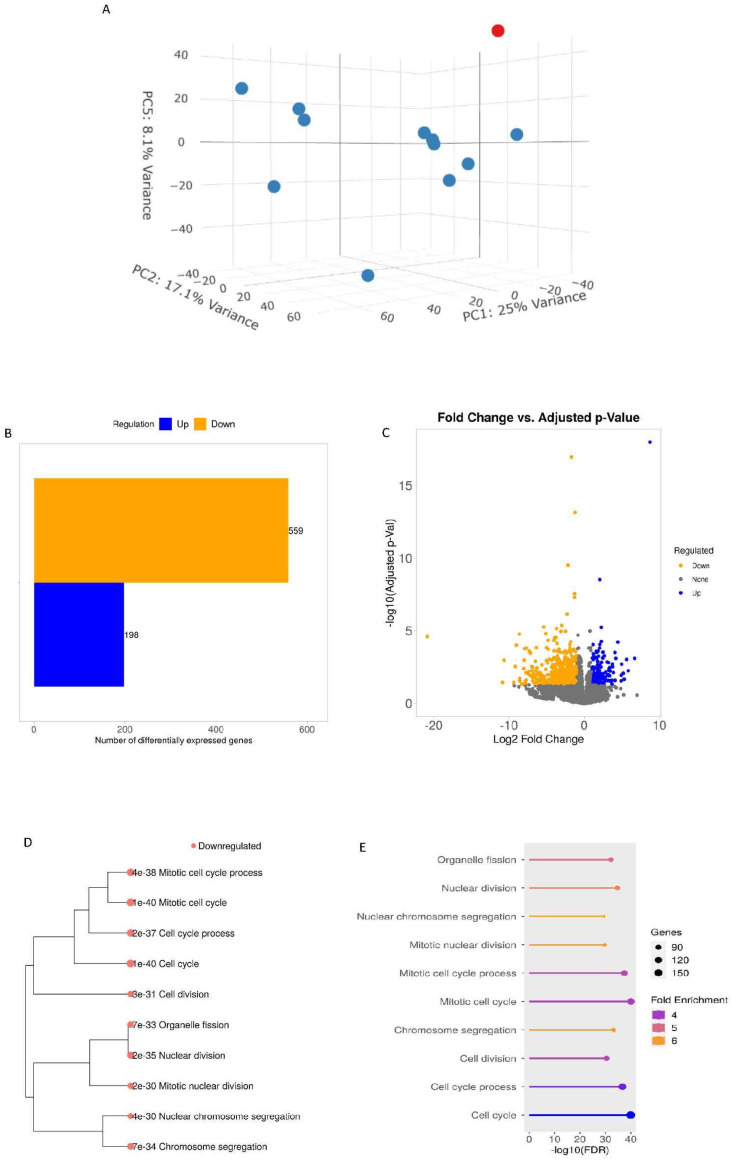
Transcriptomics analysis. (**A**) 3D-PCA analyses. The red dot indicates patient 2, and the blue dots indicate the controls. (**B**) Numbers of differentially expressed genes (DEGs) from patient vs. control comparisons; (**C**) Volcano Plot applying a cut-off of a false discovery rate (FDR) ≤ 0.05 and a log2 fold change of 2.0. Visualization of the relationship among enriched GO terms in the Biological Processes (BP) category based on down-regulated DEGs (**D**) hierarchical clustering tree with Adj.Pval and (**E**) lollipop plot where the length of the bar indicates −log10 (FDR), the color indicates fold enrichment, and the size of the circle at the tip indicates the number of genes involved.

## Data Availability

The data presented in this study are available on request from the corresponding author due to privacy/consent reasons. In the original images of the Western blot in the Appendix A section the names of samples have been covered due to privacy reasons. The original images can be obtained by contacting the corresponding author.

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
