# Peer review of "A De Novo DNM1L Mutation in Twins with Variable Symptoms, Including Paraparesis and Optic Neuropathy"

_biomolecules, 2025, doi:10.3390/biom15091230_

Round 1
Reviewer 1 Report
Comments and Suggestions for Authors
In the Materials and Methods section, several important citations and clarifications are missing:
Section 2.3: RNA-seq Analysis
- The authors should include an appropriate citation for the Salmon pipeline.
- The reference for iDEP 2.0.1 is incorrectly cited as [15]; it should be [24].
- As iDEP is a web-based tool, the authors should also indicate the date it was last accessed.
- A citation for the DESeq2 package should be included.
- The Gene Ontology (GO) analysis requires both a proper citation and the date the web resource was last accessed.
Section 2.4: Immunoblot
- The authors should provide a citation for the ImageJ (Fiji) software used in their analysis.
Section 3.2: Genetic Analysis
- The version number and citation for the Integrative Genomics Viewer (IGV) should be added.
Author Response
Reviewer 1- Comments and Suggestions for Authors.
In the Materials and Methods section, several important citations and clarifications are missing:
Section 2.3: RNA-seq Analysis
The authors should include an appropriate citation for the Salmon pipeline.
The reference for iDEP 2.0.1 is incorrectly cited as [15]; it should be [24].
As iDEP is a web-based tool, the authors should also indicate the date it was last accessed.
A citation for the DESeq2 package should be included.
The Gene Ontology (GO) analysis requires both a proper citation and the date the web resource was last accessed.
Section 2.4: Immunoblot
The authors should provide a citation for the ImageJ (Fiji) software used in their analysis.
Section 3.2: Genetic Analysis
The version number and citation for the Integrative Genomics Viewer (IGV) should be added.
Response
Thank you to Reviewer 1 for taking the time to review this manuscript and for your comments and suggestions. I have included the citations as requested for all the sections indicated:
Section 2.3: RNA-seq Analysis
- Salmon pipeline – citation n.25 “Patro R, Duggal G, Love MI, Irizarry RA, Kingsford C. Salmon provides fast and bias-aware quantification of transcript expression. Nat Methods. 2017 Apr;14(4):417-419. doi: 10.1038/nmeth.4197. Epub 2017 Mar 6. PMID: 28263959; PMCID: PMC5600148.”
- The reference for iDEP 2.0.1 now is number 26; last accessed in February 2025.
- Citation for the DESeq2 package was included - citation n.27 ”Love MI, Huber W, Anders S. Moderated estimation of fold change and dispersion for RNA-seq data with DESeq2. Genome Biol. 2014;15(12):550. doi: 10.1186/s13059-014-0550-8. PMID: 25516281; PMCID: PMC4302049.”
- Citations for the Gene Ontology (GO) analysis were included - citation n.28and 29 “The Gene Ontology Consortium, The Gene Ontology Resource: 20 years and still GOing strong, Nucleic Acids Research, Volume 47, Issue D1, 08 January 2019, Pages D330 D338, https://doi.org/10.1093/nar/gky1055” and “Steven Xijin Ge, Dongmin Jung, Runan Yao, ShinyGO: a graphical gene-set enrichment tool for animals and plants, Bioinformatics, Volume 36, Issue 8, April 2020, Pages 2628–2629, https://doi.org/10.1093/bioinformatics/btz931”. Last accessed in February 2025.
Section 2.4: Immunoblot
- the ImageJ (Fiji) software- citation n.30 “Schindelin J, Arganda-Carreras I, Frise E, Kaynig V, Longair M, Pietzsch T, Preibisch S, Rueden C, Saalfeld S, Schmid B, Tinevez JY, White DJ, Hartenstein V, Eliceiri K, Tomancak P, Cardona A. Fiji: an open-source platform for biological-image analysis. Nat Methods. 2012 Jun 28;9(7):676-82. doi: 10.1038/nmeth.2019. PMID: 22743772; PMCID: PMC3855844.”
Section 3.2: Genetic Analysis
-The version number and citation for the Integrative Genomics Viewer (IGV) were added.
IGV_2.16.0 and citation n.32 “Helga Thorvaldsdóttir, James T. Robinson, Jill P. Mesirov, Integrative Genomics Viewer (IGV): high-performance genomics data visualization and exploration, Briefings in Bioinformatics, Volume 14, Issue 2, March 2013, Pages 178–192, https://doi.org/10.1093/bib/bbs017”
Reviewer 2 Report
Comments and Suggestions for Authors
In this manuscript entitled “A de novo DNM1L mutation in twins with variable symptoms, including paraparesis and optic neuropathy”, the authors identified heterozygous DNM1L variant in monozygotic twins with differing clinical manifestations, including spastic paraparesis and optic neuropathy. They showed that the variant altered DRP1 processing and disrupted mitochondrial and peroxisomal morphology in patient-derived fibroblasts. By performing transcriptome analysis, they found that organelle fission-related pathways were dysregulated by DNM1L variants. These findings provide valuable insights into phenotypic spectrum of DNM1L-related disorders. Following are the concerns that the authors should address to improve this paper.
Specific comments
- In figure 3A and 3B, I suggest that the authors add a magnified image of the mitochondrial network to better illustrate the differences in mitochondrial morphology.
- I recommend that the authors add the citation of an article (Le et al. PMID: 39756584) to the paragraph describing DRP1 proteins in the Introduction.
- Please carefully check and correct the multiple typos and grammatical errors that occurred throughout the manuscript.
Author Response
Reviewer 2 - Comments and Suggestions for Authors.
In this manuscript entitled “A de novo DNM1L mutation in twins with variable symptoms, including paraparesis and optic neuropathy”, the authors identified heterozygous DNM1L variant in monozygotic twins with differing clinical manifestations, including spastic paraparesis and optic neuropathy. They showed that the variant altered DRP1 processing and disrupted mitochondrial and peroxisomal morphology in patient-derived fibroblasts. By performing transcriptome analysis, they found that organelle fission-related pathways were dysregulated by DNM1L variants. These findings provide valuable insights into phenotypic spectrum of DNM1L-related disorders. Following are the concerns that the authors should address to improve this paper.
Specific comments
- In figure 3A and 3B, I suggest that the authors add a magnified image of the mitochondrial network to better illustrate the differences in mitochondrial morphology.
- I recommend that the authors add the citation of an article (Le et al. PMID: 39756584) to the paragraph describing DRP1 proteins in the Introduction.
- Please carefully check and correct the multiple typos and grammatical errors that occurred throughout the manuscript.
Response
Thank you to Reviewer 2 for taking the time to review this manuscript.
For the specific comments:
- In response to the suggestion, we have included digitally magnified images of the mitochondrial network and peroxisomal morphology in Figures 3A and 3B. While these inserts were added to aid clarity, we believe the original images already convey the morphological differences effectively. The magnified regions appear somewhat grainy and may not substantially enhance the overall clarity. If acceptable to the reviewer, we would prefer to retain the original images without the inserts; however, we are also happy to keep the revised version if the reviewer finds the magnified insets beneficial.
- We added the article (Le et al. PMID: 39756584) to the Introduction.
- I hope I have corrected all the typos and grammatical errors.
Reviewer 3 Report
Comments and Suggestions for Authors
A de novo DNM1L mutation in twins with variable symptoms, including paraparesis and optic neuropathy.
I have read the manuscript with interest, and you can find my appraisal, with suggestions and concerns, section by section as follows:
Introduction: The section is well-written and informative. I have only a few suggestions. In the lines between 30-36/37, I suggest being more specific about the description of the mitochondria. Please improve the description. Moreover, I suggest adding more about the clinical cases that you reported. Indeed, it is very interesting, but you need to specify why in twins.
Methods: In my opinion, the methods are well organized, allowing a possible replication of the study.
The description of the two cases is well written, and I have to say, interesting. I am curious about a possible assessment of cognitive functions and psychiatric comorbidities, since they were not mentioned, and it would be interesting to read this information. For instance, an assessment of executive functions, processing speed, etc. Moreover, it should be interesting to see if they show some alteration of the mood or behavioral alterations. Despite this, I have appreciated the rationale that was behind each of the described steps that allow the reading to be easy to follow. In line 274, “Pt2 and control samples”, this statement is not clear, and you need to describe it better. It is not clear what you mean by control samples. Please, report in the caption of Figure 2 and Figure 3 the appropriate statistical test value, with df, and then, the p values (t(df)= value p= value). The figures are good and explanatory.
In the discussion, the results are integrated with the literature. In the conclusion, I agree with the critical view about the fact that “ genetic data alone are often insufficient”.
Author Response
Reviewer 1- Comments and Suggestions for Authors.
A de novo DNM1L mutation in twins with variable symptoms, including paraparesis and optic neuropathy.I have read the manuscript with interest, and you can find my appraisal, with suggestions and concerns, section by section as follows:
Introduction: The section is well-written and informative. I have only a few suggestions. In the lines between 30-36/37, I suggest being more specific about the description of the mitochondria. Please improve the description. Moreover, I suggest adding more about the clinical cases that you reported. Indeed, it is very interesting, but you need to specify why in twins.
Methods: In my opinion, the methods are well organized, allowing a possible replication of the study.The description of the two cases is well written, and I have to say, interesting. I am curious about a possible assessment of cognitive functions and psychiatric comorbidities, since they were not mentioned, and it would be interesting to read this information. For instance, an assessment of executive functions, processing speed, etc. Moreover, it should be interesting to see if they show some alteration of the mood or behavioral alterations. Despite this, I have appreciated the rationale that was behind each of the described steps that allow the reading to be easy to follow. In line 274, “Pt2 and control samples”, this statement is not clear, and you need to describe it better. It is not clear what you mean by control samples. Please, report in the caption of Figure 2 and Figure 3 the appropriate statistical test value, with df, and then, the p values (t(df)= value p= value). The figures are good and explanatory. In the discussion, the results are integrated with the literature. In the conclusion, I agree with the critical view about the fact that “genetic data alone are often insufficient”.
Response
Thank you to Reviewer 3 for taking the time to review this manuscript, for your suggestions, and your appreciation.
In the introduction, we expanded the description of the mitochondria, and we have now specified why reporting on twins is relevant: being monozygotic, they represent a unique opportunity to explore phenotypic variability in individuals with the same genetic background, highlighting the possible influence of additional genetic, epigenetic, or environmental modifiers. Our case descriptions to reflect the differences and similarities between the two patients. Patient 1 presented in her late twenties with severe, bilateral but asymmetric optic neuropathy and long-standing psychiatric manifestations, including depressive psychosis and obsessive-compulsive behaviors, later developing progressive spastic paraparesis. Patient 2, in contrast, showed a slowly progressive mild spastic paraparesis beginning in her early thirties, without any visual impairment or psychiatric symptoms. Regarding the reviewer’s request for information on cognitive functions and psychiatric comorbidities, formal neuropsychological testing was not performed. However, psychiatric manifestations were present in Patient 1, as described above, and were managed with antipsychotic medication. Neither patient displayed overt cognitive deficits upon standard neurological examination.
For the sentence on line 274, we clarify the concept with this new sentence “To evaluate the overall effect of the variant on gene expression, we performed a principal component analysis (PCA). The results showed a clear separation between Pt2 and control samples, indicating a distinct transcriptomic signature in the patient. This distinction was particularly evident in the 3D-PCA plot based on the expression of significantly altered genes.”
Now we reported in the caption of Figure 2 and Figure 3 the appropriate statistical test value with df. In Figure 2 for statistical analysis we used t-test (t(df)=3) **p < 0.01. In Figure 3 a total of 44 field images from four healthy controls (10 for each) and 10 field images from patient2 were used for the analysis of the mitochondrial network. For statistical analysis we used t-test (t(df)=52) ***p < 0.001. A total of 2540 objects from 3 images of the healthy control and a total of 2844 objects from 3 images of the patient2 were used for the analysis of peroxisome morphology. For statistical analysis we used t-test (t(df)=4) **p < 0.01.
Reviewer 4 Report
Comments and Suggestions for Authors
The study by Nasca A.et al is a rare example of integrating NGS sequencing technologies with functional studies on cell level. The study analyses twin sister’s genomic disorders associated with developing a variable clinical presentation, including paraparesis and optic neuropathy. Clinical exome sequencing and bioinformatics analysis, coupled with phenotypic filtering based on functional disorders optic atrophy and gait disturbance revealed mutation in DNM1L gene coding dynamine-1-like protein (DRP1), a key regulator of mitochondrial fission. A result of comprehensive genetic analysis of the pedigree was the presence of a de novo mutation in two twins identified as p.Val41Met substitution due to parental mosaicism. Moreover functional analysis of fibroblast cell culture derived from patient showing deleterious effect on gene function: altered mitochondrial morphology (swelling, dots, chain-like structures) and larger alongated peroxisomes (Fig.3), - allowed to reclassify DNM1L mutant from Variant of Uncertain Significance (VUS) to a likely pathogenic (LP) which provides a better understanding of the impact of DNM1L variant on phenotypic findings and will update the clinical diagnosis of DNM1L-related diseases. A comprehensive transcriptomic analysis through RNA sequencing (RNA-seq) (Fig.4), aiming to assess both the potential effects on DNM1L transcript structure and abundance of p.Val41Met substitution and the global impact on global gene expression evaluated 757 differentially expressed genes (DEGs) based on the comparison of patient fibroblasts with 11 independent control fibroblast cell lines.
The dynamics of the mitochondrial network (fission, fusion, mitophagy) is attracting increasing attention from the scientific community, as it plays a decisive role in the health of cells and their stable state, and the disruption of which, as noted in the Introduction, is the basis for the development of a large number of diseases. Also the current study significantly expanded our understanding of DNM1L. The aberrant bands revealed by immunoblot in sufficient quantity in patient fibroblasts as compared to minimal presence or lack in control may reflect alterations in processing of DNM1L or even might represent products of noncanonical degradation pathways operating in mutant DNM1L, reflecting a broader dysregulation of proteostasis linked to this variant.
Nevertheless there are some comments to the authors. Left column corresponding to patient fibroblasts in graphical representation of quantitative expression of OPA1 and MFN2, typical fusion mediators, (Fig.2D) looks at least 20% lower than control one. I would like to understand why the authors interpret this as the absence of the effect of mutation on the fusion process. Perhaps this is a slight compensation effect of fusion decrease as a response to impact of fission. Line 259 revealed no evidence. Line 262 when evaluating TMPs – probably the authors meant TPMs.
The study is very important potentially aiding clinicians in future diagnoses and in the refinement of genotype-phenotype correlations. Undoubtedly it should be published.
Author Response
The study by Nasca A.et al is a rare example of integrating NGS sequencing technologies with functional studies on cell level. The study analyses twin sister’s genomic disorders associated with developing a variable clinical presentation, including paraparesis and optic neuropathy. Clinical exome sequencing and bioinformatics analysis, coupled with phenotypic filtering based on functional disorders optic atrophy and gait disturbance revealed mutation in DNM1L gene coding dynamine-1-like protein (DRP1), a key regulator of mitochondrial fission. A result of comprehensive genetic analysis of the pedigree was the presence of a de novo mutation in two twins identified as p.Val41Met substitution due to parental mosaicism. Moreover functional analysis of fibroblast cell culture derived from patient showing deleterious effect on gene function: altered mitochondrial morphology (swelling, dots, chain-like structures) and larger alongated peroxisomes (Fig.3), - allowed to reclassify DNM1L mutant from Variant of Uncertain Significance (VUS) to a likely pathogenic (LP) which provides a better understanding of the impact of DNM1L variant on phenotypic findings and will update the clinical diagnosis of DNM1L-related diseases. A comprehensive transcriptomic analysis through RNA sequencing (RNA-seq) (Fig.4), aiming to assess both the potential effects on DNM1L transcript structure and abundance of p.Val41Met substitution and the global impact on global gene expression evaluated 757 differentially expressed genes (DEGs) based on the comparison of patient fibroblasts with 11 independent control fibroblast cell lines. The dynamics of the mitochondrial network (fission, fusion, mitophagy) is attracting increasing attention from the scientific community, as it plays a decisive role in the health of cells and their stable state, and the disruption of which, as noted in the Introduction, is the basis for the development of a large number of diseases. Also the current study significantly expanded our understanding of DNM1L. The aberrant bands revealed by immunoblot in sufficient quantity in patient fibroblasts as compared to minimal presence or lack in control may reflect alterations in processing of DNM1L or even might represent products of noncanonical degradation pathways operating in mutant DNM1L, reflecting a broader dysregulation of proteostasis linked to this variant. Nevertheless there are some comments to the authors. Left column corresponding to patient fibroblasts in graphical representation of quantitative expression of OPA1 and MFN2, typical fusion mediators, (Fig.2D) looks at least 20% lower than control one. I would like to understand why the authors interpret this as the absence of the effect of mutation on the fusion process. Perhaps this is a slight compensation effect of fusion decrease as a response to impact of fission. Line 259 revealed no evidence. Line 262 when evaluating TMPs – probably the authors meant TPMs. The study is very important potentially aiding clinicians in future diagnoses and in the refinement of genotype-phenotype correlations. Undoubtedly it should be published.
Response
We thank Reviewer 4 for reviewing our manuscript and for the constructive comments and for the appreciation.
Regarding the quantitative expression of OPA1 and MFN2, we previously stated that the mutation appeared to have no significant effect on the fusion process, as the patient's values were at the lower limit of the control average. However, we acknowledge that this does not exclude the possibility of a slight compensatory response, whereby reduced fusion may occur as a result of impact on fission activity.
We have also corrected the typographical errors in the manuscript.
Round 2
Reviewer 3 Report
Comments and Suggestions for Authors
The authors have addressed my concerns